# Distribution and Molecular Identification of *Culex pipiens* and *Culex tritaeniorhynchus* as Potential Vectors of Rift Valley Fever Virus in Jazan, Saudi Arabia

**DOI:** 10.3390/pathogens10101334

**Published:** 2021-10-15

**Authors:** Saleh Eifan, Atif Hanif, Islam Nour, Sultan Alqahtani, Zaki M. Eisa, Ommer Dafalla, Alain Kohl

**Affiliations:** 1Botany and Microbiology Department, College of Science, King Saud University, Riyadh 11451, Saudi Arabia; ahchaudhry@ksu.edu.sa (A.H.); inour@ksu.edu.sa (I.N.); 438106166@student.ksu.edu.sa (S.A.); 2Saudi Center for Disease Control and Prevention (SCDC), Jazan 82722-2476, Saudi Arabia; zomar@moh.gov.sa (Z.M.E.); omerda@moh.gov.sa (O.D.); 3MRC-University of Glasgow Centre for Virus Research, Glasgow G61 1QH, UK; alain.kohl@glasgow.ac.uk

**Keywords:** *C. pipiens*, *C. tritaeniorhynchus*, Rift Valley fever virus, abundance, Jazan

## Abstract

Entomologic investigations were conducted in the Al-Darb, Al-Reath, Al-Aridah, Abuareesh, Al-Ahad, Samttah, Sabyah, Damad and Beash areas by CO_2_-baited CDC miniature light traps in the Jazan region. Vectors were identified morphologically, as well as COI gene segment amplification and sequencing. The relative abundance (RA%) and pattern of occurrence (C%) were recorded. The presence of the Rift Valley fever virus (RVFV) in pooled mosquito samples was investigated by reverse transcriptase-polymerase chain reaction (RT-PCR). *Culex pipiens* (*C. pipiens*) and *Culex tritaeniorhynchus* (*C. tritaeniorhynchus*) were found with RA% values of 96% and 4%, respectively, in the region. Significant variations in vector population densities were observed in different districts. The *C. pipiens* was found highly abundant in all districts and RA% value (100%) was recorded in the Al-Darb, Al-Reath, Al-Aridah, Samttah and Damad areas, whereas RA% values (93.75%, 93.33%, 92.30% and 91.66%) were noted in Al-Ahad, Sabyah, Abuareesh and Beash districts, respectively. RA% values for *C. tritaeniorhynchus* were recorded as 8.33%, 7.70%, 6.66% and 6.25% in Beash, Abuareesh, Sabyah and Al-Ahad areas, respectively. The pattern of occurrence for *C. pipiens* and *C. tritaeniorhynchus* was recorded as 100% and 44.4% in the region. Phylogenetic analysis of *C. pipiens* and *C. tritaeniorhynchus* exhibited a close relationship with mosquitoes from Kenya and Turkey, respectively. All mosquito samples tested by RT-PCR were found negative for RVFV. In summary, the current study assessed the composition, abundance, distribution of different mosquito vectors and presence of RVFV in different areas of the Jazan region. Our data will help risk assessments of RVFV future re-emergence in the region.

## 1. Introduction

RVFV consists of enveloped, negative-sense, single-stranded segmented (S, M, L) RNA. It belongs to the *Bunyavirales* order, *Phenuiviridae* family [1,2,3]. Human infection can occur by mosquito bites, exposure to the infected animal’s blood, tissue, body fluids or aborted fetus materials [4]. The first outbreak of RVFV was reported in the livestock population of Kenya in 1931 [5] and re-occurrence has been reported mostly in African countries. Traditionally, RVFV outbreaks were considered to be confined in African and Indian Ocean islands. Previously, RVFV outbreaks were reported in Arab countries such as Egypt, Sudan and Somalia. The first outbreak of RVFV outside Africa was reported in the year 2000 in Jazan, Saudi Arabia [6]. This outbreak raised the concerns of the global spread of RVFV at different locations such as Asia, Europe and America due to climate change, international trade of live animals and animal products. It is difficult to predict RVFV epidemics but typically a repeat of infection cycles has been observed every 8 to 10 years in African countries [7,8]. RVFV is reported to be transmitted by a variety, and different types of mosquito species. Transmission of viruses across international borders can be attributed to the import and export of livestock and related products, as well as translocation of mosquitos and human travel and transportation [4]. Potential arthropod vector identification is important to understand which mosquitoes species may maintain the virus during inter-epidemic phases. It is also important to gain a clear understanding of the community composition and abundance of such mosquito species that may act as a vector for RVFV survival and transmission [9,10]. Survival of RVFV in a specific environment greatly depends on the availability of competent vectors abundance in addition to different factors such as population density of livestock, temperature conditions and rainfall. All these factors play a key role in the breeding of vectors and the virus’s maintenance and replication [2,11,12]. The RVFV outbreak in the Jazan region of Saudi Arabia resulted in the death of 40,000 different types of animals including goats, sheep, cattle and camels. Moreover, around 10,000 abortions were recorded in different types of animals. Control measures including animal vaccination programs, aerial and land spray of insecticides for control of adult and larval stages of mosquitoes, surveillance of mosquito, detection of RVFV by molecular techniques and sero-surveillance of animals were adopted in affected areas [13]. However, a few sporadic cases were recorded in the last 10 years but no spread of the virus was recorded outside the area [6]. In the post epidemic phase, it is important to carry out surveillance on the status of virus circulation in different areas. Assessment of distribution and abundance of mosquito vector species in the affected area are also critical for the implementation and monitoring of comprehensive disease/vector control programs. Previously different mosquito species were collected, identified from the affected areas and RVFV occurrence in mosquitos was investigated. An ELISA-based study on blood samples of different types of livestock in the Jazan region stated RVFV infection rates of 0.12% in Sabya and 1.04% in Jizan districts in the year 2006. In other studies, blood samples from local and sentinel herds from the Jazan region were tested by ELISA and an RVFV infection rate of 0.15% and 0.12% was estimated. Mosquito vector abundance, distribution and RVFV circulation were investigated in the Jazan region and dominance of *Culex* species was recorded with 98.13% and 99.26% in Sabia and Abuareesh areas. Whereas all mosquito samples were found negative for the presence of RVFV [14,15,16].

Mosquito vectors are known to maintain and transmit RVFV, so data generated regarding the existence, composition and abundance of different species will help to define outbreak risk and distribution of vectors in affected areas. Understanding of maintenance and survival of RVFV during inter-epidemic periods is limited, so the current study hypothesized that RVFV is maintained in vector mosquitoes by transovarial transmission and active surveillance is essential for monitoring of RVFV circulation in the affected areas. The current study aimed to assess species composition, abundance, distribution of different mosquito vectors and the existence of RVFV in nine areas of the Jazan region. The outcomes of the study will help to identify vector populations and risk areas in the region for better monitoring and control of important veterinary and public health threats.

## 2. Materials and Method

### 2.1. Study Area

Jazan is located in the Southwest region of Saudi Arabia; it comprises a 22,000 km^2^ area with a population of 1.3 million distributed in six cities and 3500 villages. It is situated in the subtropical zone and lies between 16°–12, and 18°–25, latitude North. The region has an average rainfall of 800 mm per year and high ecological diversity, with different kinds of vegetation and animal species [17].

### 2.2. Collection of Adult Mosquito Samples

Fifty-four CO_2_ baited CDC miniature light traps were used for collection of adult mosquitoes in nine Provinces. Five villages from each Province were selected based on their locations as collection sites to cover the entire Province, the distances between the villages ranged from two to three kilometers. Every night, in each village, six CDC light traps were distributed in six shaded animals’ shelters (1 trap/shaded shelter) at 6 PM and collected at 6 AM; the minimum distances between the animals’ shelters were about 150 m; sheep and goats numbers in these shelters ranged from 30–150. Mosquitoes were also collected from three valleys, Wadi Beash in Beash Province, Wadi Jazan in Abuareesh Province and Wadi Kholab in Al-Ahad Province. Six traps were distributed along the valley, the distance between the sites was about 500 m. All the collected mosquitoes were identified morphologically and confirmed by molecular identification. Morphological identification was performed on a cold ice block using the mosquito taxonomy key [18]. Only females were pooled into groups of 10 mosquitoes/pool and stored at −70 °C until use.

### 2.3. Relative Abundance

The relative abundance (RA%) for mosquito species for nine different sites was assessed and represented in terms of ratio between specimen number of specific species and the total specimen number of all species of mosquitos captured in the area ×100 [19].

### 2.4. Pattern of Occurrence

Mosquito species distribution was assessed by means of the pattern of occurrence (C%). It was represented in terms of ratio among the number of positive sites for mosquito occurrence and total sites included in study [20].

### 2.5. Molecular Confirmation of Mosquito Identification

#### 2.5.1. DNA Extraction

Mortar and pestle were used to homogenize mosquito legs (mini borosilicate glass chamber length 60 mm/pestle diameter 9.0 mm 3.0 mL, Fisherbrand, Loughborough, UK) in 100 μL of Minimum Essential Media (MEM) (manufactured Euro Clone, UK). DNA was extracted from the homogenate using RealLine DNA–Extraction 2 (BIORON Diagnostic, Römerberg, Germany) following the manufacturer’s recommendations: 300 μL of lysis Reagent with the sorbent (magnetic particles) added to homogenate in 1.5 tubes and placed into the thermo-shaker for five minutes at 65 °C, 1300× *g*. Then 400 μL of DNA precipitation solution added to each tube and mixed for 15 s in a vortex. The samples were centrifuged (13,000× *g*) for five minutes at room temperature. The pellets were washed twice, dried at room temperature and re-suspended. The extracted DNA was stored at −86 °C.

#### 2.5.2. Amplification of Mosquito COI Fragment

A 710 bp of template was amplified using the forward primer LCO1490 and reverse primer HCO2198 [21]. PCR mix with total volume of 25 μL comprised of 12.5 μL of GoTag^®^G2 green master mix (Promega) 2 μL (20 pmol) of universal forward and reverse primers, 2 μL of DNA template and 8.5 μL nuclease-free water. Thermal cycler was programmed for initial denaturation (94 °C × 3 min), 30 cycles [denaturation (94 °C × 60 s), annealing (50 °C × 60 s, extension (72 °C × 60 s)] and final extension (72 °C × 5 min).

In each run, we used nuclease-free water as negative control and extracted DNA from *Culex pipiens* mosquitoes reared in Saudi Public Health Authority insectary in Jazan as positive controls. The PCR products amplification analyzed by gel electrophoresis (1.5% agarose in Tris-Acetate EDTA buffer), stained with ethidium bromide and viewed by Gel Doc XR Imaging System (Bio-Rad, Irvine, CA, USA).

### 2.6. Sequencing and Data Analysis

PCR products (710bp) were sequenced commercially in Macrogen Co., Ltd., Seoul, South Korea. ABI PRISM^®^ BigDyeTM Terminator Cycle Sequencing Kits (Applied Biosystems) were used to perform Sequencing reactions according to the manufacturer instructions in thermal cycler (MJ Research PTC-225 Peltier). Single-pass sequencing was performed on each template using LCO1490 forward primer. BigDye XTerminator purification protocol was used for purification of fluorescent labeled fragments from unincorporated terminators. The samples were re-suspended in distilled water and processed by ABI 3730xl sequencer (Applied Biosystems, Carlsbad, CA, USA).

The Basic Local Alignment Search Tool was used for bioinformatics analysis. The evolutionary relationship tree was created for the sequenced samples. The mosquito nucleotide sequences analysis was performed by MEGA X software [22]. A total of 29 *Culex pipiens* and 67 *C. tritaeniorhynchus* nucleotide sequences including highly divergent isolates belonging to the same species were analyzed. The alignments of sequences were performed by ClustalW on default settings (Opening penalty 15, extension penalty 6.66). Phylogenetic trees were constructed on the basis of best fit model for nucleotide substitution by utilizing the minimum Bayesian information criterion [23]. Phylogenetic tree reliability was assessed by boot strapping of 1000 replicates. The Tamura three-parameter method was used for estimation of genetic distances [24].

### 2.7. Detection of RVFV in Mosquitoes

#### 2.7.1. RNA Isolation

GeneJET Viral DNA and RNA Purification Kit from Thermo Scientific (Carlsbad, CA, USA) was used to extract RNA following the manufacturing procedure; 200 μL of lysis buffer supplemented with RNA carrier and 50 μL Proteinase K were added to 200 μL of diluted (according to manufacture instructions) in 1.5 mL tube and mixed immediately by vortexing and incubated for 15 min at 56 °C in a thermomixer. In this mixture, 300 μL of ethanol (100%) was added, vortexed and incubated at room temperature for 5 min. The lysate was transferred to a Spin Column and centrifuged 6000× *g* for 1 min. The Spin Column was transferred to a new tube, wash buffer was added and centrifuged for 1 min at 6000× *g*. The Spin Column was placed into a new collection tube and washed twice by wash buffer 2 at 6000× *g* for 1 min then centrifuged at 16,000× *g* for 3 min. The Spin Column was then placed into elution tube and 50 μL elution Buffer was added and incubated at room temperature for 2 min, then centrifuged at 13,000× *g* for 1 min.

#### 2.7.2. Reverse Transcriptase Polymerase Chain Reaction

The test was performed as described by [25]. NSca forward primer and NSng reverse primers (Table 1) synthesized by Macrogen Company (Seoul, South Korea) were used to amplify 810 basepairs of viral genome. Single-step RT-PCR was performed by following the protocol of access RT-PCR–system Thermo Fisher Scientific Inc., Waltham, MA, USA). The mixture with total volume of 50 μL comprised of 25 μL of 2X RT-PCR buffer, 1 μL of 25X RT-PCR Enzyme Mix, 1 μL (50 pmol) of NSca and NSng primers, 5 μL RNA template and 17 μL nuclease-free water. Incubation was performed at 42 °C/1 h for cDNA synthesis followed by initial denaturation (94 °C × 5 min), 35 cycles [denaturation (94 °C × 30 s), annealing (55 °C × 30 s, extension (72 °C × 60 s)] and final extension (72 °C × 5 min).

#### 2.7.3. Nested-PCR

Nested-PCR was carried out with total volume of 50 μL comprised of 25 μL of GoTag^®^G2 green master mix (Promega, Madison, WI, USA), 5 μL (1:50 dilution) RT-PCR product, 50 pmol (final concentration 1 μM) of each consensus forward primer NS2g and reverse primers NS3a (Table 1). Themal cycler was programmed for initial denaturation (94 °C × 3 min), 30 cycles denaturation (94 °C; 30 s), annealing temperature (55 °C; 30 s), primer extension (72 °C; 1 min) and final extension for 5 min. Nuclease-free water was used as negative control and RVFV cDNA of positive sample obtained from Saudi Public Health Authority Jazan was used as positive control. The nested PCR products were resolved by 2% agarose, stained with ethidium bromide and viewed by Gel Doc XR Imaging System (Bio-Rad, Irvine, CA, USA).

## 3. Results

A total of 2168 mosquito vectors were collected from the nine areas including Al-Darb, Al-Reath, Al-Aridah, Abuareesh, Al-Ahad, Samttah, Sabyah, Damad and Beash in the Jazan region (Table 2) identified by morphological characteristics, 710 bp COI gene segment amplification and sequencing. Two species *C. pipiens* (96%) and *C. tritaeniorhynchus* (4%) were found in different areas.

Strong variations were recorded in vector population densities in different districts. *C. pipiens* was found highly abundant with an RA% value of 100% in the Al-Darb, Al-Reath, Al-Aridah, Samttah and Damad areas. Abundance with RA% values 93.75%, 93.33%, 92.30% and 91.66% was recorded in the Al-Ahad, Sabyah, Abuareesh and Beash districts, respectively. *C. tritaeniorhynchus* abundance was observed with RA% values in Beash 8.33%, Abuareesh 7.70%, Sabyah 6.66% and Al-Ahad 6.25%. The pattern of occurrence for *C. pipiens* and *C. tritaeniorhynchus* was recorded as 100% and 44.4%, respectively.

The morphological identification of species was confirmed by PCR and sequencing. The evolutionary relationship of taxa confirms morphological identification and represents the relationship between Jazan species and other species from the Genbank. The phylogenetic tree of *C. pipiens*/2B64-9.20 (nucleotide sequences from the current study) showed a clear relationship (Figure 1 and Figure 2) with species identified from Kenya (*C. pipiens*/KCH9, *C. pipiens*/NEH12 and *C. pipiens*/NAH4), Portugal (*C. pipiens*/Port-2168) and Turkey (*C. pipiens*/S41) with an evolutionary distance of approx. 0.009194 (Appendix A).

The best substitution model validation has yielded the ML method and Tamura 3-parameter model [24] to best depict the evolutionary history of *C. pipiens* (Appendix A). The highest log-likelihood of the obtained tree was −1054.32. The percentage of trees in which the associated taxa clustered together is shown next to the branches. A matrix of pairwise distances estimated using the Maximum Composite Likelihood (MCL) approach (Appendix A) indicated the closest isolate identified by the smallest divergence distance. The tree is drawn to scale, with branch lengths measured in the number of substitutions per site.

Moreover, the phylogenetic tree of *C.*
*tritaeniorhynchus*/2B64-9.20 (nucleotide sequences from the current study) displayed a close relationship (Figure 3 and Figure 4) to a species identified from Turkey (*C. tritaeniorhynchus/MBIM1-A3*) with an evolutionary distance of approx. 0.00464 (Appendix A).

The best substitution model validation has yielded the ML method and Tamura 3-parameter model [24] to best depict the evolutionary history of *C. tritaeniorhynchus* (Appendix A). The highest log-likelihood of the obtained tree was −1118.55. The percentage of trees in which the associated taxa clustered together is shown next to the branches. A matrix of pairwise distances estimated by using the Maximum Composite Likelihood (MCL) approach (Appendix A) indicated the closest isolate identified by the smallest divergence distance. A discrete Gamma distribution modeled the evolutionary rate differences among sites (5 categories (+G, parameter = 0.0500). The rate variation model allowed for some sites to be evolutionarily invariable ([+I], 48.00% sites). The tree is drawn to scale, with branch lengths measured in the number of substitutions per site.

All mosquito samples collected from different areas of the Jazan region were tested by RT-PCR and nested PCR was found negative for RVFV (Table 3) compared to the Smithburn vaccine strain vaccine result (Figure 5).

## 4. Discussion

Entomological surveillance was performed to assess the role of different types of mosquitos as RVFV carriers in nine districts of the Jazan region of Saudi Arabia. Genetic diversity and distribution of mosquito vectors were studied by using DNA-based markers provided useful information regarding the composition and abundance of vector species in these areas. These findings can be used to assess the risk of future RVFV re-emergence. The mosquito surveillance identified the existence of *C. pipiens* and *C. tritaeniorhynchus* species of mosquito in nine districts of Jazan. A total number of mosquitos captured by CO_2_ traps upon identification showed 96% belonged to *C. pipiens* and 4% belonged to *C. tritaeniorhynchus* species. These findings are comparable with a previous study where *Aedes vexans arabiensis*, *C. pipiens* and *C. tritaeniorhynchus* species were found in abundance in the same region [26]. *C. pipiens* can act as a carrier to transmit RVFV in humans and animal hosts whereas *C. tritaeniorhynchus* were confirmed as carrier vectors of RVFV during the first outbreak in the year 2000 in Jazan and their coexistence as the carrier can increase the risk of an RVFV outbreak and its spread [27,28,29]. *C. pipiens* was also found with high abundance in Okavango Delta Botswana [30].

*C. pipiens* was reported as an important vector responsible for RVFV outbreaks in East Africa [31]. *C. pipiens* was also described as the main vector for the distribution of RVFV in Egypt and the Magreb region [28,32]. On the other hand, *C. tritaeniorhynchus* distribution was reported in the central and southwestern part of the Albaha region [33] and *C. tritaeniorhynchus* prevalence with niche modeling was established for the RVFV-affected region of Jazan [34]. The difference in the distribution pattern of mosquito species in different areas may correspond to the habitat and ecology of different areas [35]. In a previous study, different types of mosquito genera including *Culex* (91.73%) and *Aedes* (22.13%), as well as potential vectors such as *Phlebotomus* (15.73%) and *Culicoides* (6.4%) were identified in Jazan [16]; in the current study we recorded the presence of *C. pipiens* and *C. tritaeniorhynchus* in different areas and the abundance of *C. tritaeniorhynchus* was lower as compared to previous reports [16,27].

Population densities of *C. pipiens* and *C. tritaeniorhynchus* may be relevant and higher densities of *C. pipiens* populations in territories of the Jazan region might be linked with a risk of future RVFV outbreaks. Indeed in a previous study, *Culex* species were found to be linked with RVFV outbreaks in the region [27]. The current study depicted the existence of mosquito vectors species in nine areas of the Jazan region with distinct habitats and ecological conditions. These data can be utilized for risk assessment of future RVFV outbreaks along with other risk factors. Different studies documented the existence of RVFV vectors in Turkey and *C. pipiens* was reported as most abundant (26.7%) followed by *C. tritaeniorhynchus* species (23.8%) [36,37]. The data obtained from the Jazan region also showed the presence of two *Culex* species in abundance, which may impact RVFV epidemic patterns. *C. pipiens* was recorded as the most abundant species in the Jazan region. These findings are in line with the abundance of *C. pipiens* in different regions of Kenya [38]. This could be owing to spatial considerations since Kenya is located in the most eastern part of Africa, which is close to the Southern region of Saudi Arabia.

After the year 2000, limited data were gathered on mosquito vectors abundance, distribution and RVFV circulation during the silent period in affected areas, and only one study reported dominance of *Culex* species with 98.13% and 99.26% abundance with negative PCR results in the Sabia and Abuareesh areas of Saudi Arabia [27]. Our study depicted the abundance and composition of mosquito vector species in nine areas of the Jazan region. These findings along with other ecological factors will help to determine the risk of RVFV future outbreaks. Alongside taxonomic identification of mosquito vectors, DNA markers were also utilized to assess genetic diversity and distribution. The phylogenetic analysis of *C. pipiens* (*C. pipiens*/2B64-9.20) and *C. tritaeniorhynchus* (*C.tritaeniorhynchus*/2B64-9.20) exhibited a relationship with mosquito species from Kenya and Turkey. The surveillance data from 11 provinces of Turkey analyzed by DNA barcoding and COI gene sequencing exhibited the occurrence of *C. tritaeniorhynchus* with high genetic differentiation [39]. *C. pipiens* are known to transmit and spread different arboviruses, for example, the West Nile virus, which was first reported in Uganda in 1937. The existence of *C. pipiens* has already been verified around the world in the Middle East, Central Asia, Africa, Europe and the American continent [40]. The arrival of mosquito species closely related to those of the East African region in Saudi Arabia may occur by crossing through the narrow waterway between the Gulf of Aden and the Red Sea by strong winds or air currents or by transportation and trade channels.

All vector samples collected from the different areas of the Jazan region were found negative for the presence of RVFV. This finding is in line with previous studies in the same region [16] where no RVFV was detected in mosquito vectors. In an entomological investigation in 2012 in the Ngorongoro region of Tanzania, 1823 mosquitoes were grouped in pools of 5 to 10 mosquitoes, tested for the presence of RVFV during an inter-epidemic period and no positive sample was identified [41]. This may correspond to the low level of circulation or absence of the RVFV in the study area but raises the question of where and how the virus is maintained. Moreover, control measures such as mass vaccination of ruminants, eradication of infected livestock, import ban on livestock and products from East Africa, awareness programs and surveillance of the area resulted in the efficient control of outbreaks [6].

## 5. Conclusions

The current study demonstrated the existence of *C. pipiens* and *C. tritaeniorhynchus,* as well as their distribution and abundance in nine areas of the Jazan region. Both vectors are known to carry, maintain and transmit RVFV, so their presence may indicate the risk of future outbreaks. Although our study was concerned with the abundance, distribution and diversity of mosquitoes as vectors for RVFV, other factors can be also attributed to virus transmission including vectorial capacity, vector competence and survival along with the environment and ecological parameters. Mosquito vector surveillance, distribution and competence evaluation are key factors for the assessment and prediction of risk. The surveillance and distribution patterns of vectors will help to improve disease prevention strategies including insect vector control.

## Figures and Tables

**Figure 1 pathogens-10-01334-f001:**
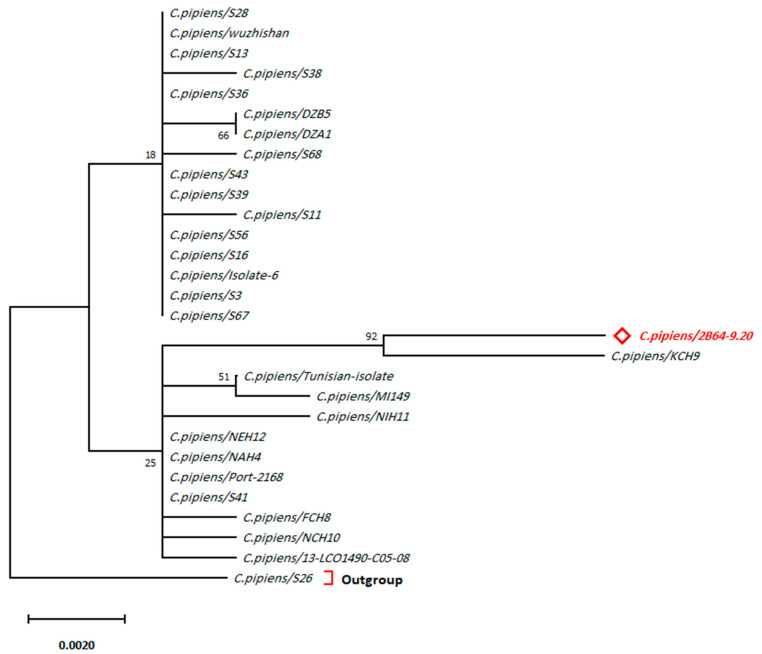
Evolutionary analysis by Maximum Likelihood (ML) method of *C. pipiens.* Geographical locations of sequences are mentioned in Appendix A. Red bold text indicates the sequence from the current study. Outgroup refers to the highest divergent sequence belongs to the same species.

**Figure 2 pathogens-10-01334-f002:**
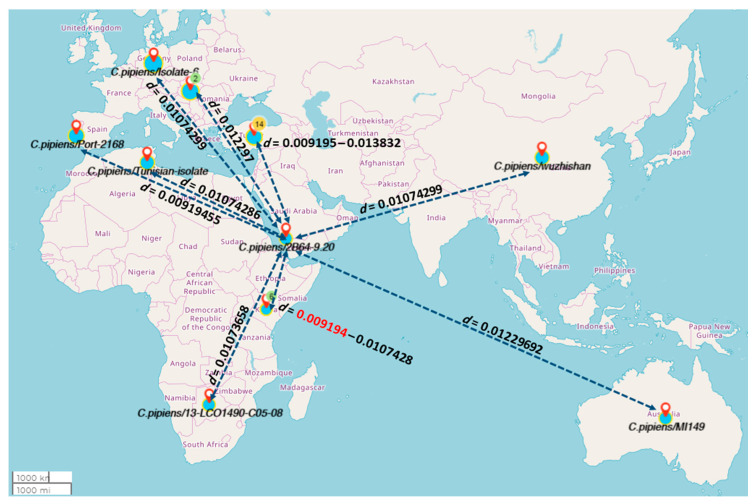
Map showing the evolutionary relationship of closer sequences to the Saudi Arabian *C. pipiens* sequence (2B64-9.20). *d* denotes the pairwise evolutionary distance. The red bold number refers to the smallest divergence distance. The numbers in circles at any country refers to the count of sequences derived from this country, thus an evolutionary distance range might be mentioned at this country.

**Figure 3 pathogens-10-01334-f003:**
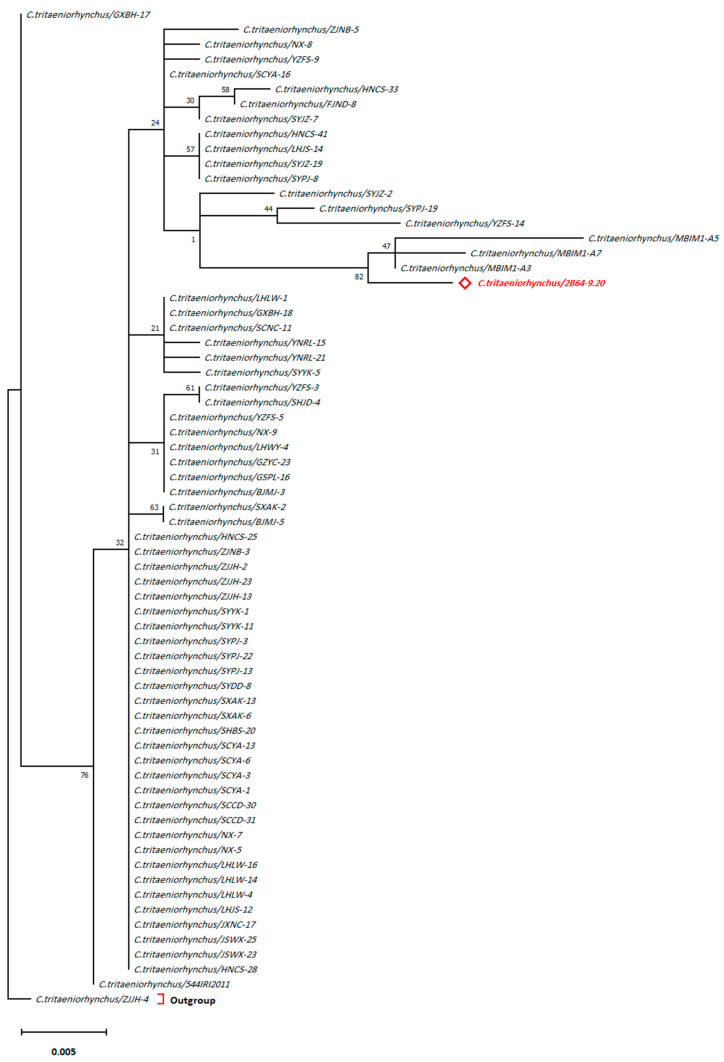
Evolutionary analysis by ML method of *C. tritaeniorhynchus.* Geographical locations of sequences are mentioned in Appendix A. Red bold text indicates the sequence from the current study. Outgroup refers to the highest divergent sequence belongs to the same species.

**Figure 4 pathogens-10-01334-f004:**
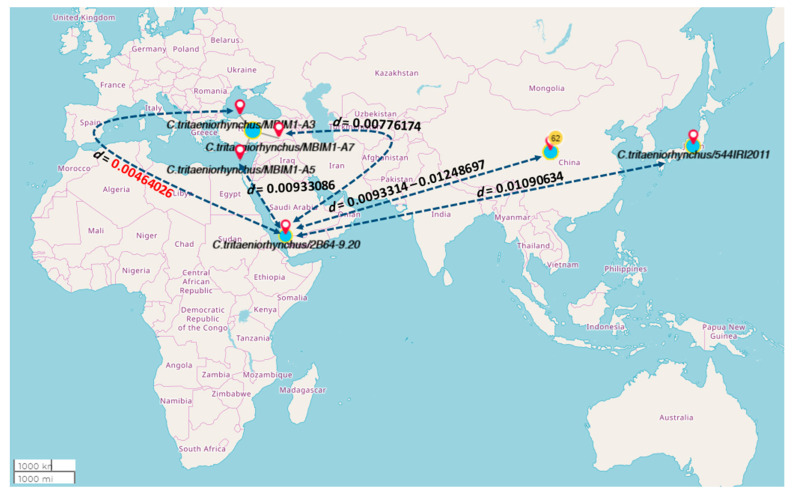
Map showing the evolutionary relationship of closer sequences to the Saudi Arabian *C. tritaeniorhynchus* sequence (2B64-9.20) denotes the pairwise evolutionary distance. The red bold number refers to the smallest divergence distance. The number in yellow circles in China refers to the number sequences used from this country, thus a range of evolutionary distance is mentioned in this country.

**Figure 5 pathogens-10-01334-f005:**
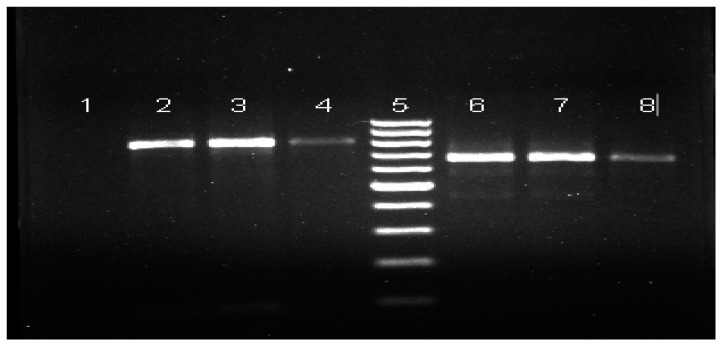
Agarose gel electrophoresis showing RT-PCR and Nested PCR of RVFV. Lanes 1–4: amplified 810 bp band using NSca, NSng primers (RT-PCR product), Lane 1: negative control, Lane 2: positive control, Lanes 3 and 4 RVFV vaccine lane, Lane 5: 100 bp DNA ladder. Lanes 6–8: amplified 667 bp DNA (Nested PCR), Lane 6: positive control, Lane 7 and 8 RVFV vaccine.

**Table 1 pathogens-10-01334-t001:** Primers used for detecting RVF virus.

Primer	Sequence	Amplicon Size (bp)
NSca	5′ CCTTAACCTCTAATCAAC 3′	810
NSng	5′ TATCATGGATTACTTTCC 3′	
NS2g	5′ GATTTGCAGAGTGGTCGTC 3′	667
NS3a	5′ ATGCTGGGAAGTGATGAGCG 3′	

**Table 2 pathogens-10-01334-t002:** Number of mosquitoes and abundance of *Culex* species in the various districts.

District	*C. pipiens*			*C. tritaeniorhynchus*		
	Number of Mosquitoes	RA %/Species *	RA %/Total Mosquitoes **	Number of Mosquitoes	RA %/Species *	RA %/Total Mosquitoes **
Al-Darb	298	100	58.33	0	0	0.00
Al-Reath	103	100	35.71	0	0	0.00
Al-Aridah	173	100	47.06	0	0	0.00
Abuareesh	252	92.30	44.44	21	7.70	3.70
Al-Ahad	390	93.75	53.57	26	6.25	3.21
Samttah	193	100	55.56	0	0	0.00
Sabyah	322	93.33	50	23	6.66	3.57
Damad	163	100	50	0	0	0.00
Beash	187	91.66	44	17	8.33	4.00

*: Relative abundance from two studied mosquitoes species; **: Relative abundance from all other mosquitoes species.

**Table 3 pathogens-10-01334-t003:** RT-PCR results of RVFV in tested mosquitoes.

Districts	Al-Darb	Al-Reath	Al-Aridah	Abuareesh	Al-Ahad	Samttah	Sabyah	Damad	Beash
No of samples	14	5	8	13	16	10	15	7	12
RVF positive	0	0	0	0	0	0	0	0	0

## Data Availability

The data presented in this study are available in the main text, figures, tables and Appendix A.

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
