# Peer review of "Distribution and Molecular Identification of Culex pipiens and Culex tritaeniorhynchus as Potential Vectors of Rift Valley Fever Virus in Jazan, Saudi Arabia"

_pathogens, 2021, doi:10.3390/pathogens10101334_

Round 1

Reviewer 1 Report

Review of Pathogens-1367613

Rift Valley Fever virus (RVFV) is a major arbovirus threat in Africa and the Middle East, particularly to livestock. This manuscript presents an analysis of the phylogenetic relationships among two mosquito vectors of Rift Valley Fever virus (RVFV) in Saudi Arabia and the outcome of RVFV surveillance from the mosquitoes. A reasonable number of Culex pipiens and Cx tritaeniorhynchus were collected and COI genes sequences to establish phylogenies. Attempts were made at RVFV virus detection by PCR however no detections were made. While this work is important and the the mosquito collections represent a lot of effort, there are several shortcomings to the manuscript. In particular, the reason for including the phylogenetic analysis is not adequately stated and little is discussed of these trees and their significance. The lack of virus detection is unsurprising as the sample size is relatively small for the purposes of detecting virus from mosquitoes during an inter-epidemic period.

Main comments

Line 77-82. More detail needs to be provided to describe the tapping procedures. Line 78 includes “six CDC light traps distributed daily in animal shelters and valleys”.  How many animal shelters and valleys was this (were these different animal shelters and valleys?). How widely were the shelters/valleys separated? Can you describe the typical location of trap placement (e.g. indoors or outdoors, shaded?).How many trapping nights were conducted at each location?

The authors state that there have only been sporadic cases of RVF cases in the 10 years prior to this study. Did any disease cases occur in the sampled regions during the course of this study? This is important information for interpreting the virus detection results (See below).

Table 2 I’d expect to see more detail from these data. It would be useful to include the counts of mosquitoes obtained and not just relative percentages.

Figure 1 includes data from all study sites collapsed into single measures of occurrence and relative abundance for all study sites. It may be useful to be able to see the variation across the six study sites from this figure. The authors have provided these values for the different districts in the paragraph. Can the authors combine all the data into Figure 1 or Table 2?  

Lines 110-11.  Please describe the positive and negative controls.

Lines 163-164. Again, please describe what you are referring to as “positive and negative controls”

Figure 4. The highlighted evolutionary distance with Kenya (d= 0.009194) is very similar to distances to sequences from Spain (d= 0.00919455) and Turkey (d=0.009195). Are the estimated distances between these three locations statistically significantly different (is there a test?) If this cannot be tested, I suggest discussing Spain and Turkey together with Kenya as locations that share the smallest distance in the Results and Discussion.

Figures 3 and 5. These figures provide phylogenetic relationships among Culex pipiens and Culex tritaeniorhynchus. Each sample is labelled by a the species name and a code (not described). Clusters are present but these are not described or discussed further in the manuscript. The sample name codes are not intuitive or informative, and therefore the figures have limited value to the average reader. I suggest that additional annotation is included to these figures to identify whether the clustering is related to other factors (for example geographic location). If geography is determining the clusters, the sample names can include a name specifying the sample collection location. Additionally, vertical lines could be drawn beside clusters that are also labelled by their sample collection location. Furthermore, the main features of Figures 3 and 5 should be presented in the text in Results and discussed in the Discussion section. Pages of Evolutionary Divergence estimates are provided in Supplementary Material. Can the authors discuss what their data indicate about the history of these mosquito species in Saudi Arabia?

The authors present data showing zero detections of Rift Valley Fever virus from mosquito pools. Table 3 provides the results from 100 mosquito pools (of 10 mosquitoes each = 1000 mosquitoes), collected from 9 locations. This unfortunately represents a relatively small sample for a study aiming to detect virus in mosquitoes during an inter-epidemic period. The prevalence of arbovirus infections in mosquitoes during inter-epidemic periods can be extremely low (>1%). The prevalence during epidemics increases if mosquitoes are targeted from affected regions. 22% of 105 pools of mosquitoes collected during an epidemic in Kenya were positive for RVFV (LaBeaud et al. Emerg Infect Dis. 2011 Feb; 17(2): 233–241). Seven isolates of RVFV only were obtained from 1,287 pools (61,347 mosquitoes) in Kenya during an earlier outbreak (Logan et al. J Med Entomol 1991 Mar;28(2):293-5). As this study occurred during an inter-epidemic period, the prevalence of infection in mosquitoes would have been substantially lower that observed in this studies. I don’t believe that the sample size in this study provides sufficient power to detect a low circulating rate of RVFV infection in mosquitoes in an inter-epidemic period. RVFV may be circulating and the lack of a detection may be due to the small sample.

The authors tested pools of mosquitoes. Did the authors test the affect of mosquito pooling on virus detection (versus testing individual mosquitoes), or was this tested in the paper that was cited for the methods? Were DNA samples diluted? Pooling can increase the concentration of PCR inhibitors. Ideally controls should also include pools of mosquitoes, spiked by a single positive mosquito or spiked with virus isolate. Controls were used, which is reassuring, however little detail is provided about what these were. These points are important because no virus was detected from any mosquito pool. The authors should comment on the likelihood that the negatives are false negatives.

It should be mentioned that mosquito species abundance and distribution are not the only factors incriminating them as vectors of an arbovirus. The other components of the Vectorial Capacity model, including mosquito vector competence and survival are major contributing factors.

Minor comments

Line 45. The statement is made “Transmission of virus mostly depends on competent species of mosquito.” This sentence is vague (are the authors discussing RFV here or viruses in general). The previous sentence states that RFV is transmitted by mosquitoes, therefor this sentence can be deleted.

Lines 48-50. The statement is made “Potential arthropod vectors identification is important which are known to maintain the virus during inter- pandamic phases. It is important to gain a clear understanding regarding the RVFV hiding reservoirs during the silent periods.” The term reservoir is typically used to describe vertebrate hosts, not mosquitoes (which act as vectors). Vertebrates are the principle ‘reservoirs’ for RVFV.

Line 49-50  “Inter-pandamic” (misspelled) presumably should be “inter-epidemic”

L56-58. Please correct the grammar in this sentence

L74-75. Can the authors please provide more detail in their description of the study site. Altitude? Climate? Geography? Limit to 2-3 sentences. See my comments about annotation of figures 3 and 5. If there are factors that determine clusters in these regions such as location or geopgraphy, then describe these factors here.

Line 147. Remove ‘the’ from “as described by the [20]”

Line 151. Change “RT_PCR” to “RT-PCR”

Lines 163-164. Again, please describe what you are referring to as “positive and negative controls”

Figure 2 is unnecessary; the results can be more efficiently described in the text.

Lines 214-226. Some of the writing in this section is a description of how the phylogenetic analysis was conducted, not Results. These sentences should be transferred to Material and Methods.

Figures 3 and 5. In phylogenetic trees, the outgroups selected were sequences from the same mosquito species as for the other samples in trees. It is my understanding that outgroups are normally selected from more distantly related species; however, I am not a specialist in phylogenetic analysis. Please make specific mention of the two outgroup sequences used in the relevant section in Materials and Methods and provide a brief justification for why these sequences were used.

Author Response

Dear Respected Reviewer,

Thanks a lot for offering time for revising our manuscript. Your valuable comments were highly appreciated. Kindly, find the "Reviews 1 comments and author replies" file attached for your consideration.

Best regards,

Authors

Reviewer 2 Report

Review report

Manuscript ID: pathogens-1367613
Type of manuscript: Article
Title: Distribution, molecular identification of Culex pipiens and and Culex 
tritaeniorhynchus, as potential vectors of Rift Valley fever virus in Jazan, 
Saudi Arabia  

Authors: Saleh Eifan, Atif Hanif, Islam Nour, Sultan Alqahtani, Zaki M. 
Eisa, Omar M. Alhassanand, Alain Kohl

Special Issue: Bunyavirus

Broad comments: 

Bunyaviruses (e.g. Rift fever virus, Crimean-Congo hemorrahagic fever virus, Akabane virus, Schmallenberg virus, Hantaan virus, La Crosse virus and much others) are worldwide pathogens, and cause of serious diseases to humans and/or animals. Bunyaviruses are recognized as posing an increasing threat to human health and are good examples of cause of ‘emerging infections’.

It is essential to monitor vectors of these pathogens, to guide preventive and control measures.

Although this is an interesting subject, the present article is not able to achieve the aim of the special issue. In addition, the article is not clear, some statements are confuse, needing moderate editing of English language and style, and additional information is needed to improve the paper. The authors didn’t explain why the research they have done is important for the scientific community and for the local policy makers in terms of control and preventive measures. They didn’t identify the hypotheses and why they are important to address. The article does not reads well as a whole, there is a gap between the introduction, the materials and methods, the results and then the discussion. The text is very repetitive.

Specific comments:

Title:

Minor change:

Title: ‘Distribution, molecular identification of Culex pipiens and and (…)’ – remove and

Authors should be consistent: Abbreviations should be defined in parentheses the first time they appear in the abstract, main text, and in figure or table captions and used consistently thereafter.

Abstract:

All abstract and keywords: Italic - Culex pipiens and Culex tritaeniorhynchus (including italic in the abbreviated form).

Line 12: CO2 – should be CO2

Line 14: RA% - and remove all the spaces thereafter (all text)

Lines 15-16: the sentence does not makes sense – re-write.

Lines 20-22: this sentence also does not makes sense the way is written.

Line 22: For C. pipiens a 100% pattern of occurrence was recorded, whereas C. tritaeniorhynchus showed 44.4% pattern of occurrence.

Line 24: mosquito

Lines 27-28: The data obtained will help to improve the disease control and spread strategies. – What do you mean with spread strategies?

  1. Introduction

Minor changes:

Authors should be consistent – first time in the text, should be Culex pipiens and Culex tritaeniorhynchus, then use the abbreviated form C. pipiens and C. tritaeniorhynchus

Lines 33-34: [1] [2] – should be [1,2]

Lines 38-39:

Minor changes: In recent past outbreaks of RVFV are reported in Arabian Peninsula like Egypt, Saudi Arabia and Yemen.  – This sentence should be re-writen.

Line 39: remove the before Africa

Line 51: the same with the refs [8] [9]

Lines 56-57: the sentence should be re-writen.

Line 58: remove The before control measures

Lines 62-63: this sentence is repeated – see lines 58-59

Lines 64-65: The sentence should be re-writen and Implement should be implement

Line 71: threats, and remove the before important

Major changes:

The authors refer: Previously different mosquito species were collected, identified from the affected areas and RVFV occurrence in mosquitos was investigated [12,13].  Was this in Jazon? Ok, so what were the conclusions from these previous studies and where this study is going to be different, more informative, innovative? Is this point going to be raised in the discussion?

This is not well addressed later in the discussion – the reader does not understands what are the differences between the previous and the current studies in Jazan. The introduction should be re-writen, whith the hypotheses of the study being well indentified.

  1. Materials and methods

Minor changes:

Lines 79-80: All the collected mosquitoes were used morphological and molecular identification.  – re-write this sentence.

Line 93: Each mosquito legs homogenized in a mortar and pestle  - re-write

Line 107: Themal – should be Thermal

Line 117: Sequencing should be sequencing

Line 140: 6,000 × g – remove the spaces

  1. Results

Minor changes:

Line 175: remove The

Lines 175-181: re-write these sentences

Line 181: figure 1 should be Figure 1

Line 184: figure 2 should be Figure 2

Lines 196-202: some of this information should be in the materials and methods. The same for the information in lines 214-226

Discussion

Major changes:

Points to address in the discussion:

Novelty of the molecular tecniques used, if any?

What are the implications that C. pipiens is more abundant than C. tritaeniorhynchus?

Why is this interesting and in what way this can help understanding the outbreaks in Jazan? ‘The phylogenetic tree of C. pipiens/2B64-9.20 (Nucleotide sequences from current study) showed the clear relationship (Figures 3, 4) with species identified from Kenya’

I think the maps showing the evolucionary relationship of Culex species are great. Could this be better explored in the discussion?

Minor changes:

Line 243: Genetic diversity

Lines 243-245: not sure if the sentence makes sense. Check English

‘Genetic diversity and distribution of vectors were studied by using DNA based markers, and provided useful information to map the risk areas for future RVFV outbreaks. ‘

Lines 245-246: C. pipiens and C. tritaeniorhynchus -use the abbreviated form

Line 249 – needs a full stop before These

Line 249: These findings are compareable with a previous study where

Line 250 – again, use the abbreviated form of Culex – please change all text in the discussion using the abbreviated form.

Line 251: ‘dance in the same region [21]’.

Line 255: ‘RVFV in Egypt and Magreb region [24] [25]. ‘ – This is not the way to add the refs – should be [24,25].

Line 260: In a previous study

Line 262: Jazan; In the current study

Line 264: was

Line 277: C. pipiens – the same in lines 281/282 for the 2 species

Line 283: mosquito

Line 288: region [12] where no RVFV was detected in mosquito vectors.

Line 291: remove space between and surveillance

  1. Conclusion

Lines 294/295: C. pipiens and C. tritaeniorhynchus  - use the abbreviated form  

Author Response

Dear reviewer,

Thanks a lot for offering time for revising our manuscript. Your valuable comments were highly appreciated. Kindly, find the "Reviews 2 comments and author replies" file attached for your consideration.

Best regards,

Authors

Round 2

Reviewer 1 Report

Thank you for addressing my comments. These address most of my concerns.

Table 2. The RA% for  C. pipiens and C. tritaeniorhynchus from each location sum to 100%. This indicates that no other Culex or other species were collected. I would like to have seen the abundance  of C. pipiens and C.tritaeniorhynchus shown relative to all mosquitoes (from all species) that were trapped, not just the relative abundance among these two species only. Use a more informative column heading than “mosquitoes”. Do you mean number of mosquitoes?

As I mentioned in my first review, it would be helpful if the authors added annotation on figures 1 and 3 to improve interpretation of the clustering. If geography is a major driver of the clustering, then the authors could add labelled brackets to the right of the tree that show which sequences are from the same geographic region.

Please include more information in Figure 1 and 3 legends. E.g. describe why labels are red.

Some minor grammatical errors remain.

Author Response

Dear Respected Reviewer,

Thanks a lot for offering time for reviewing our paper for the second time and four your valuable comments. Kindly, find attached "Revision 2_Reviewer 1 comments and author replies".

Best regards,

Authors

Reviewer 2 Report

Dear authors, 

Although I think the discussion is still a bit repetitive in some parts, the article has improved greatly. 

Some minor points

Introduction. Lines 72-79 and 88-95 are repeated.

Line 115 – CO2

Lines 349-351 – substitute 'indeed' in one of the sentences.

Lines 377 – whuch – change for which

The discussion is just one paragraph! – please do paragraphs in the discussion.

Author Response

Dear Respected Reviewer,

Thanks a lot for offering time for reviewing our paper for the second time and four your valuable comments. Kindly, find attached "Revision 2_Reviewer 2 comments and author replies".

Best regards,

Authors
